# On the Role of TATA Boxes and TATA-Binding Protein in *Arabidopsis thaliana*

**DOI:** 10.3390/plants12051000

**Published:** 2023-02-22

**Authors:** L. K. Savinkova, E. B. Sharypova, N. A. Kolchanov

**Affiliations:** Federal Research Center, Institute of Cytology and Genetics, Siberian Branch of Russian Academy of Sciences, 10 Prospekt Akad. Lavrentyeva, Novosibirsk 630090, Russia

**Keywords:** TATA box, TATA-binding protein, TATA box and environment, adaptation, TBP, plant morphology

## Abstract

For transcription initiation by RNA polymerase II (Pol II), all eukaryotes require assembly of basal transcription machinery on the core promoter, a region located approximately in the locus spanning a transcription start site (−50; +50 bp). Although Pol II is a complex multi-subunit enzyme conserved among all eukaryotes, it cannot initiate transcription without the participation of many other proteins. Transcription initiation on TATA-containing promoters requires the assembly of the preinitiation complex; this process is triggered by an interaction of TATA-binding protein (TBP, a component of the general transcription factor TFIID (transcription factor II D)) with a TATA box. The interaction of TBP with various TATA boxes in plants, in particular *Arabidopsis thaliana*, has hardly been investigated, except for a few early studies that addressed the role of a TATA box and substitutions in it in plant transcription systems. This is despite the fact that the interaction of TBP with TATA boxes and their variants can be used to regulate transcription. In this review, we examine the roles of some general transcription factors in the assembly of the basal transcription complex, as well as functions of TATA boxes of the model plant *A. thaliana*. We review examples showing not only the involvement of TATA boxes in the initiation of transcription machinery assembly but also their indirect participation in plant adaptation to environmental conditions in responses to light and other phenomena. Examples of an influence of the expression levels of *A. thaliana* TBP1 and TBP2 on morphological traits of the plants are also examined. We summarize available functional data on these two early players that trigger the assembly of transcription machinery. This information will deepen the understanding of the mechanisms underlying transcription by Pol II in plants and will help to utilize the functions of the interaction of TBP with TATA boxes in practice.

## 1. Introduction

Development of multicellular organisms and the uniqueness of each cell are implemented through the expression of certain genes at the right time in the right place. This phenomenon requires a complicated modulation of gene expression by regulatory factors interacting with specific DNA sequences and among themselves; these processes are critical for the growth, development, and survival of organisms. The foundation of this phenomenon is the fact that, while the number of protein-coding genes has remained fairly constant throughout the evolution of metazoans, the number of regulatory DNA elements has increased dramatically [1]. The modulation of transcription via fine-tuning the rate and timing of initiation by RNA polymerase II (Pol II) may be key to the alteration of organism-wide gene expression patterns in response to developmental and environmental cues [2]. Thus, elucidating the regulation of gene expression requires a detailed knowledge about the mechanisms of transcription.

In eukaryotes, transcription is carried out by three DNA-dependent RNA polymerases. Pol I and Pol II produce ribosomal RNAs and messenger RNAs for ribosome assembly and protein synthesis, respectively. Pol III specializes in synthesizing 5S rRNA, transfer RNAs (tRNAs), U6 spliceosomal RNA (U6 snRNA), and small noncoding RNAs, including 7SL RNA, 7SK RNA, and RNase P RNA [3]. It should be noted that in plants, two more plant-specific RNA polymerases—RNA polymerase IV and RNA polymerase V—have been identified that synthesize noncoding RNAs necessary for transcription and for gene repression via the RNA-directed methylation of DNA [4]. The promoter region of the genes that are read by Pol II is usually defined as a functional region located upstream of the transcription start site and containing specific DNA sequences and regulatory elements. Promoters can be up to a thousand base pairs long. In promoters, a so-called core promoter is thought to be present: a minimal DNA sequence occupying a region of ~50–100 nucleotides (the length depends on the type of promoter) upstream and downstream of the transcription start site. The core promoter is necessary for accurate positioning of transcription machinery relative to the starting nucleotide of transcription [5,6].

The core promoter is the primary site for the assembly of the preinitiation complex [7,8]. It contains the transcription start site (TSS) at position +1 and approximately 50–100 bp upstream and downstream for control over the precise initiation of transcription [8]. The core promoter used to be regarded as a universal component, functioning identically for all protein-coding genes. It is now well established that core promoters differ in their architecture and function and that each core promoter is quite unique [9,10]. A core promoter may contain one or more short DNA sequences (called core promoter elements or motifs) that contribute to its function. A DNA element called an initiator (Inr) is most often present in the region of transcription initiation in eukaryotic core promoters. Inr is not a very conserved element, containing A and C in the transcription start site (−1; +1) and surrounded by several pyrimidines [11]. There are other highly common core promoter elements located downstream of the transcription start site: MTE is a 10-nucleotide motif situated before a downstream core promoter element (DPE) between positions +18 and +27 [12] and the DPE downstream of the transcription start site at +28 to +33 bp [13], which Juven-Gershon et al. consider the main promoter element in *Drosophila* development-related genes [6,9,14]. *Bridge* is a rare element of a core promoter; it contains the first and third subregions MTE and DPE and is the site of interaction of TAF6 and TAF9 subunits of TFIID (transcription factor II D) [15,16]. TFIID is a key factor for transcription initiation from class II promoters and is composed of TBP and TBP-associated factors that are thought to bind to the DPE; the cooperative binding of TFIID requires certain precise distances between the Inr and the DPE [15]. The MTE and DPE are conserved from *Drosophila* to humans and their functions depend on the presence of a functional Inr; they are enriched in TATA-free promoters [16,17]. Figure 1 shows the layout of some elements of the eukaryotic core promoter. XCPE1 and XCPE2 (X core promoter element 1 and X core promoter element 2) direct TFIID-independent transcription by Pol II from TATA-free promoters of mammals [18,19].

Recognition of the core promoter by transcription machinery is necessary for proper positioning and assembly of the preinitiation complex. Biochemical studies have shown that TBP binds to a TATA element in both directions [20]. It follows from this finding that the correct orientation of the assembly of the transcription complex on a promoter is determined by other promoter elements. For instance, an element called BRE (TFIIB recognition element) was discovered as a sequence that promotes the highly specific binding of TFIIB to a complex of human or archeal TBP with a TATA box [21]. BRE consists of upstream and downstream sequences (BRE_u_ and BRE_d_, respectively) [21,22], which are located upstream and downstream of a TATA box. The binding of TFIIB to these sequences stabilizes the TBP–TATA box complex and this stabilization is absolutely necessary for the subsequent binding of Pol II to it [23] and for preinitiation-complex assembly. Numerous studies have shown that TFIIB functions as a single polypeptide having a two-domain structure. The C-terminal domain of TFIIB (cTFIB) contains an imperfect direct-repeat motif [24] and the N-terminal domain of TFIIB (nTFIIB) contains a zinc-rich region followed by a TFIIB region highly conserved among many species [25]. Crystallographic analysis of a complex of cTFIIB with a TBP–TATA complex has revealed that direct repeats of cTFIIB are responsible for the interaction with TBP [24]. Aside from interacting with DNA upstream and downstream of a TATA box associated with TBP, cTFIIB interacts with the “wall” region of yeast RNAP II in preinitiation complexes [26]. The crystal structure of the ternary complex TFIIB–TBP–TATA obtained at 2.7 Å resolution indicates that core TFIIB recognizes the TBP–TATA complex via protein–protein and protein–DNA interactions. The N-terminal domain of core TFIIB forms the ternary complex’s lower surface that can fix a transcription start site [24]. By crystallographic analysis, it has also been demonstrated that TFIIB positions the coding strand of DNA in the active site of Pol II; this arrangement also ensures the correct choice of a transcription start site [27]. Thus, the role of TFIIB in the formation of the preinitiation complex cannot be overestimated.

The architecture and composition of the core promoter determine the initiation of transcription—starting with the assembly of the preinitiation complex specific for this type of promoter—and are responsible for the implementation of enhancer–promoter specificity (so-called preferential activation), thereby predetermining the course of the subsequent stages of transcription, such as the promoter release, Pol II arrest, or the transition to productive elongation. The core promoter also participates in the regulation of termination, polyadenylation, RNA polymerase recycling, and the modulation of translation [5,28]. Different types of core promoters are differentially recognized by preinitiation complexes containing or lacking TBP [5].

## 2. Structure of TBP

TBP is a subunit of a general (basal) transcription factor called TFIID. It is known that the most important function of TBP in TBP- and TFIID-containing preinitiation complexes is promoter DNA bending [29] (Figure 2), which makes it possible for TFIIB to fix the bent conformation of the TBP–TATA subcomplex [30,31,32], thus enabling the landing of Pol II in the complex with TFIIF. In contrast to other transcription factors, which are not necessary for promoters if the latter do not contain their binding sites, TBP is required for the transcription of all TFIID-dependent promoters, including those lacking a TATA box [32].

Obviously, on TATA-containing promoters, the DNA–protein interaction plays a decisive role, whereas on TATA-free promoters, the specificity of the interaction of TBP is implemented via mechanisms of direct and indirect interaction with activator proteins and TBP-associated factors. On a TATA-free but TFIID-dependent promoter, the positioning of TBP near nucleotide position −30 is mediated by the interaction of TBP-associated factors with other core promoter elements, by the interaction of TFIIB with BRE, and by protein–protein interactions of TBP and of TBP-associated factors with activators [16,17,34].

Until recently, it has been believed that TBP is a general transcription factor necessary for the transcription of all genes. Today, three paralogs of TBP are already known; these are so-called TBP-related factors: TRF1, TRF2, and TRF3. All of them share homology with the core domain of TBP. TRF1 and TRF3 can interact with a TATA box, but TRF2 cannot because it has lost the phenylalanine residue that intercalates into DNA [35]. The core promoter element with which it interacts has not yet been identified [36]. All three paralogs of TBP initiate assembly of specific preinitiation complexes on promoters of certain groups of genes responsible for embryonic development, differentiation, and morphogenesis [36,37].

Analysis of the amino acid sequence of TBP of humans, of *Drosophila*, of yeast, and of *Arabidopsis* has shown that it consists of a phylogenetically conserved carboxyterminal domain 180 amino acid residues long and a variable amino-terminal domain [38]. The C-terminal domain of TBP is approximately 80% identical among proteins from different species [33,38] and can autonomously direct efficient and specific transcription in vitro when other general transcription factors and Pol II are added [39]; therefore, it is sometimes referred to as the core domain. The N-terminal domain of TBP varies in length and structure among different organisms; it is thought to ensure species-specific interactions of TBP [40,41].

Crystallographic analysis of full-length TBP of *Arabidopsis thaliana* has revealed that the C-terminal domain of eukaryotic TBP has a pseudo-symmetrical structure formed by ten β-folds (S1–S5 and S1′–S5′, which constitute a slightly curved anti-parallel β-sheet resembling a concave DNA-binding saddle) and by four α-helices (H1, H2, H1′, and H2′, situated on the upper side of the molecule) [42].

The bending of the *AdML* promoter carrying a TATA box in a complex with yeast or human TBP has been demonstrated in original experiments involving an electrophoretic gel shift assay [40]. In 1993, crystal structure was determined for complexes of the C-terminal domain of *S. cerevisiae* TBP with the TATA box of the *CYC1* promoter [43] and of full-length TBP of *A. thaliana* with the TATA box of the *AdML* promoter [44]. It was found that the binding of saddle-shaped TBP to 8 bp TATA box TATAAAAR causes a DNA bend that forms an 80–100° angle between strands of the DNA leaving the complex. In this context, the TATA box is kinked in two places (the so-called “two kink model”) [45] between the first and second and between the seventh and eighth base pairs of the TATA box. In the nascent complex, the minor groove—with which the β-sheet of the concave surface of TBP interacts and intercalates four phenylalanine residues into DNA (two on each side)—is expanded and partially unwound. The concave side of the saddle performs specific binding to DNA, whereas the convex side interacts with transcription factors [43,44]. The unique conformation of the −30 region of DNA is required for preinitiation complex assembly and correlates with transcription efficiency in vitro and in vivo [46].

Unlike most DNA-binding proteins, which specifically interact with the major groove of DNA to form hydrogen bonds between amino acid residues and nucleotides, TBP when choosing a target interacts with the minor groove of DNA and recognizes structural features such as flexibility or deformability [47]. The specificity of the binding of TBP is determined to a large extent by the recognition of the DNA structure because, due to the symmetrical arrangement of donor and acceptor groups in the minor groove of DNA, it is almost impossible to distinguish between AT and TA (and between GC and CG), whose combination constitutes a TATA box.

## 3. Structure and Variation of the TATA Box

The eukaryotic core promoter element that was identified first (in *Drosophila* histone genes) and is best known is the TATA box [48] with the TATAWAAS consensus sequence (where W = A or T, whereas S = G or C according to the IUPAC nomenclature), to which TBP binds and initiates assembly of the preinitiation complex. Although the TATA box is conserved from yeast to humans, it has been detected only in a minority of core promoters in almost all these species: 20% to 46% of core promoters in yeast [49], ~10% in humans (24% together with TATA-like sequences) [50], 64% of promoters in *Drosophila* [51], 27% of promoters in mice [52], and ~39% of promoters in *A. thaliana* [53] (or 29% according to the first genome-wide study by Molina et al. [54]), whereas only ~19% of promoters in rice contain a TATA box [55]. Studies on yeast and humans indicate that genes containing the TATA box are usually subject to tissue-specific expression and are mainly regulated by stressful stimuli, whereas genes without TATA are expressed constitutively and are predominantly involved in housekeeping processes [56]. After a meticulous analysis of experimentally proven promoters of eukaryotic genes, Bucher [57] constructed a position weight matrix (Table 1), whose elements were expressed in logarithmic units. So far, this work has not lost its relevance as a generally accepted way to identify a TATA box in eukaryotic promoters.

In addition to the table format, frequency matrices are presented as a picture: a sequence logo (Figure 3). Each column of the logo corresponds to a position in the matrix and the height of each letter is proportional to the frequency of the letter at this position.

In the Bucher frequency matrix, T at the third position has the greatest weight (frequency of occurrence); next, As at the second and fourth positions have the same weight; As at the fifth and sixth position have less weight.

In contrast to the frequency matrix of mammals, in the frequency matrix of plants, the greatest weight belongs to A at the fourth position (Table 2). Somewhat lower frequency of occurrence belongs to the first three nucleotides TAT as well as to A at the sixth position [58,59]. As readers can see, except for slight variations in the frequency of each base, the consensus sequences in plants and mammals are very similar.

The literature data indicate wide variations of the TATA box sequence among natural promoters. Different researchers give dissimilar estimates of the prevalence of TATA boxes. For example, it is reported in Ref. [49] that ~24% of promoters in the human genome contain TATA-like elements and only 10% of them contain the canonical TATA box. The authors of Ref. [17] report 10–20% and the authors of Ref. [60] report 11–17%. Cooper et al. [61] believe that only approximately 10–15% of TATA-containing promoters of class II genes in mammals have the canonical TATA box. Instead of the canonical TATA box sequence, other promoters contain sequences with 1–2 substitutions in the consensus, which are called TATA-like elements [62].

It has been shown in yeast that expression regulation fundamentally differs between genes containing the TATA consensus and genes containing a TATA-like element. Genome-wide studies indicate that transcription of approximately 10% of the yeast genome is predominantly regulated by SAGA and 90% of the genome by TFIID. Most SAGA-dependent yeast genes are stress-induced and tightly regulated, whereas TFIID-dependent yeast genes are mostly housekeeping genes [63]. SAGA-dependent genes of yeast contain a consensus TATA box with the sequence TATAWAWR, whereas TFIID-dependent ones used to be considered TATA-free; however, in 2012, some investigators [62] showed that 99% of these “TATA-less” promoters contain a TATA-like element that carries two or fewer substitutions relative to the TATAWAWR consensus [62]. Since then, there has been growing evidence of differences in regulation, but still only in yeast [64,65,66]. Baptista et al. [67] have re-examined the participation of SAGA in global transcription in *S. cerevisiae* and demonstrated that it is the general transcription factor that is recruited to most promoters of genes transcribed by Pol II, where SAGA plays a critical role in mRNA synthesis regardless of the promoter type (SAGA- or TFIID-dominated, TATA-containing, or TATA-free). In this regard, SAGA can be compared to the mediator, which is required for the whole transcription by Pol II and stimulates the formation of the preinitiation complex. The authors of Ref. [67] concluded that the question about the function of SAGA in transcription in higher eukaryotes is still open. In higher eukaryotes, as a rule, TATA-containing (and, accordingly, TFIID-dependent) genes are inducible but are not housekeeping genes. It is very likely that in higher eukaryotes, the regulation of transcription of genes directed by TATA-like elements also has its own specific features. There is currently no validated definition of a TATA-like element in higher eukaryotes.

Although the TATA box (and TATA-like elements) is usually only one of several synergistically acting core promoter elements ensuring precise positioning of TBP on a promoter, TATA-containing promoters can be highly sensitive to substitutions in their sequence. It is known that, for different promoters, identical substitutions in the TATA box have dissimilar effects on promoter activity [57,68,69].

The interaction of TBP with a TATA-like element largely depends on flanking sequences. Flanking-sequence motifs affecting the TBP–TATA interaction are unique to certain TATA boxes [70]. It is reported that the information content of the flanks also depends on the type of TATA box: symmetrical, as in *E4* (TATATATA), or asymmetrical, as in *AdML* (TATAAAAG) [70].

A high information content may point to the presence of functional binding sites for other transcription factors, e.g., BRE, in the flanks of a TATA box [70].

Results implying an effect of flanking sequences on the affinity of oligonucleotides for TBP are presented in Ref. [69]. Experimental determination of equilibrium dissociation constants of complexes of TBP with oligodeoxyribonucleotides has shown that the affinity of TBP for the oligonucleotides that differ in the content of AT pairs in the sequences flanking the TATA box varies by a factor of 25–30 [71].

## 4. Assembly of the Preinitiation Complex

It has been traditionally believed that, on TATA-containing promoters, the assembly of the preinitiation complex is triggered by the interaction of TBP, which is a component of the general transcription factor TFIID, with the TATA box. This sequential assembly model was described in 1994 [72] and can be outlined roughly as follows. TBP within TFIID interacts with the TATA box, and the resulting TBP–TATA complex, in which the −30 region of the promoter DNA is bent, serves as a substrate for recognition by subsequently arriving general transcription factors, the first of which are TFIIA and TFIIB. By binding, TFIIB in turn recruits Pol II in the complex with TFIIF to the promoter. The entry of TFIIE and TFIIH into the complex completes the formation of the preinitiation complex. According to the sequential assembly model, for basal transcription, TBP alone is sufficient for the recognition of the TATA box and for subsequent binding of other general transcription factors. On the contrary, TBP alone is not enough for implementing regulated transcription. The latter requires TFIID, which consists of TBP and TBP-associated factors (Figure 4).

Although the TATA box is regarded as an element capable of directing transcription independently of an Inr, experimental evidence [73] suggests that even for the *AdML* promoter containing the consensus TATA box (sequence TATAAAAG), the DCE (downstream core element) [74], and the Inr, when Inr is replaced by an element called TCT (polypyrimidine initiator), the TFIID binding does not occur.

Another pathway of preinitiation complex assembly is assumed to be mediated by the recruitment of a preassembled holoenzyme complex of Pol II, which comprises Pol II, general transcription factors (such as TFIIB, TFIIE, TFIIF, and TFIIH), and proteins partaking in other cellular functions [75].

The large number of proteins required for transcription has always raised the question of spatial organization of transcription in the nucleus and how quickly the factors required at a given moment can be delivered. Ultra-high-resolution microscopy of live cells has revealed the dynamic foci of Pol II, which are termed concentrates or condensates [76,77,78,79]. It is believed that these temporary foci arise via liquid–liquid phase separation of proteins having disordered regions [80]. The phase separation is based on multivalent and cooperative interactions between intrinsically disordered protein regions; this phenomenon concentrates proteins in the cell [81,82]. Condensates emerge around transcription factors and help to deliver proteins for transcription initiation to sites of high transcriptional activity [83]. Temporary condensates participate in various processes, including RNA metabolism, ribosome biogenesis, the DNA damage response, signal transduction, and others.

## 5. Comparative Structural Analysis of Core Promoters of *A. thaliana*

Research into the structural characteristics of DNA shows that, for promoters of mammals (*Mus musculus* and *Homo sapiens*), the GC content (%) exceeds the AT content in all regions except for the region from −30 to −26 bp, where these percentages are almost equal. In contrast, *A. thaliana* core promoters contain the highest AT content in all regions, peaking between positions −35 and −26 bp [84]. Promoter sequences of *Drosophila melanogaster*, *Caenorhabditis elegans*, and *Danio rerio* contain approximately equal AT and GC percentages, whereas the region −31 to −26 bp is also enriched with nucleotides A and T [84]. According to the same authors, in core promoters of mammals, of *D. melanogaster*, *D. rerio*, and of *C. elegans*, the occurrence of TATA reaches a maximum at position −28 or −29 bp, whereas in *A. thaliana*, this maximum is at −31 bp relative to a transcription start site. The maximum of occurrence of AAAA is at position −26 bp in mammals, *D. melanogaster*, *D. rerio*, and *C. elegans* and at −28 bp in *A. thaliana*. The distribution of dinucleotides and tetranucleotides in the region of a transcription start site varies fairly widely among species. Position +1 is mostly occupied by A in almost all species [84]. It is known that the TATA box and its variants are characteristic of tissue-specific genes possessing a focused promoter (mostly having either one transcription start site or a strongly dominant one among several), whereas the transcription of housekeeping genes, which represent 60–70% of vertebrate genes, can be initiated from many transcription start sites (a dispersed promoter) [85].

*A. thaliana* genes possessing a TATA box also tend to have a sharper dominant peak of a transcription start site than other genes [86], which have a broader peak. Mammalian dispersed promoters typically contain CpG islands [87], upon which a transcription factor called Sp1 lands [88]. Unlike mammalian promoters containing CpG islands, plant promoters are enriched in Y-patches: TC-rich regions that were first identified in the genome of several plants in an analysis of a local distribution of short sequences (LDSS) [89,90]; ~18% of *A. thaliana* promoters and 50% of rice promoters contain TC regions that can be regarded as analogs of mammalian CpG islands [56,91,92]. These TC regions are considered a class of regulatory elements for transcription under specific conditions [91]. The TC elements seen in plants are absent in both humans and mice [91]. TATA-containing genes are usually induced by stress signals and their expression is usually inhibited during growth, when genes not containing TATA boxes gain an advantage in expression [20,93,94]. Genes with TATA-free promoters are usually housekeeping genes [31]. It should be noted that data on the distribution and localization of core promoter elements in plants, in particular in *Arabidopsis*, are rather inconsistent due to the technical shortcomings of the methods used in the past. For instance, the authors of Ref. [95], which is the first genome-wide study on the prevalence of core promoter elements in eight plant genomes (four dicotyledons, which include *Arabidopsis*, and four monocots), have documented a statistically significant difference (with a large number of false positive results) in the prevalence of each element, except for the TATA box, DPE, and Y-patches. Nucleosome-free DNA in the region of core promoters of six studied species of multicellular eukaryotes has properties that are necessary for structural changes that promote the binding of TBP and the formation of an open promoter complex [95]. These properties include the ease of TATA box melting, the ability to expand the DNA minor groove and to bend toward the major groove, and the rigidity of nucleotides flanking the TATA box.

## 6. Mutations in TATA Boxes of *A. thaliana*

There are very few experimental studies on the interaction of TBP with TATA elements in promoters of plant genes, in particular in *A. thaliana*, compared with genes of yeast, *Drosophila*, humans, and other mammals. For example, Mukumoto et al. [96] have investigated in detail the effect of substitutions in *A. thaliana* TATA boxes on in vitro transcription in a *HeLa* nuclear extract, where human TBP is replaced by TBP from *A. thaliana* (Table 3). Those authors showed that the replacement of the third T by A in the sequence of TATA box T_1_A_2_T_3_A_4_T_5_A_6_T_7_A_8_ diminishes transcription by 85%, whereas other substitutions, A to T or T to A, preserve more than 36% of the transcription activity seen with the original TATA box. Three TATA box sequences (TATATATA, TATAAATA, and TATATAAA), which are effective in the transcription of eukaryotic genes, were also effective in transcription with either *A. thaliana* or human TBP. In that study, substitutions from A or T to C or G at position 2, 3, 4, or 5 strongly reduced transcriptional efficiency, whereas substitutions at position 1, 6, or 7 had a weaker impact. Among the mutant TATA elements, sequences TAGAGATA and GAGAGAGA resulted in the complete suppression of transcription.

The authors of Ref. [97] have researched the influence of mutations in a TATA box having the sequence TATAAATA on the mRNA expression of the *CaMVsynT-3* gene in *A. thaliana* protoplasts and found that, if this sequence is replaced with either TAcgAATA or TATAcgTA, the gene expression is inhibited by 95% (Table 4). After replacement of TATAAATA by cgTAAATA, up to 7% of the expression remained and, when the TATAAAcg sequence was tested, 27% of the initial activity (registered with TATAAATA) was retained. The sequences flanking the TATA box and the distance to a transcription start site were not altered in these experiments. Similar findings [88,89] indicate the importance of positions 3 and 4 in the TATA box for transcription.

Yamaguchi et al. [98] have performed the first detailed analysis of TATA sequence requirements for in vitro transcription initiation by Pol II using tobacco as an example. They also investigated transcriptional activity of the rice wild-type *PAL* promoter having the TATA box sequence T_1_A_2_T_3_A_4_T_5_A_6_T_7_A or its mutant versions in tobacco, *Drosophila*, and HeLa transcription systems. Although the relative transcriptional activity varied among these promoters, those authors reported that all three systems produce similar results, pointing to the functional conservatism and similarity of TATA box sequence requirements for in vitro transcription among plants, insects, and humans (Figure 5).

Nonetheless, if one takes a closer look and tries to quantify the presented data, one will notice substantial differences in transcription among these three eukaryotic species after identical mutations of the TATA box. For clarity, we present the data of the diagram [98] as Table 5.

For TATA box No. 4 in the tobacco transcription system, the activity is 20% of that of the consensus sequence TATATATA; in the HeLa system it is 40%; in *Drosophila* it is 10% (a two-fold difference) (Table 5). For TATA box No. 8, the activity in tobacco is 7%, in HeLa 17%, and in the *Drosophila* system no transcription is observed at all with this TATA box. For TATA box No. 10, in tobacco the activity is ~7%; in HeLa it is >30%; in *Drosophila* there is no activity at all. For TATA boxes No. 12 and 13, in tobacco the activity is <5%, which corresponds to the maximum weight of this letter in Shahmuradov’s plant weight matrix [58]. Accordingly, its substitution has a dramatic impact on transcription efficiency in the tobacco and fly systems but not in HeLa (in Drosophila, 0% of activity and in tobacco, <5% of activity, respectively). A possible reason for such dissimilarities is differences in the composition of nuclear extracts owing to interspecies differences. These data also show that, for stand-alone point mutations in the TATA box, the most crucial nucleotides in *A. thaliana* are T’s at the third (TATA boxes No. 8 and 10) and fifth positions (TATA boxes No. 15 and 16) and A at the fourth position (TATA boxes No. 12 and 13) and at the sixth position (TATA boxes No. 17–19), in good agreement with the Bucher and Shahmuradov frequency matrices [57,58]. Among the mutant elements, two multiple-base substitution variants, TAGAGATA and GAGAGAGA, proved to be completely ineffective at initiating transcription in all three cell types. It must be mentioned that when distances from the TATA boxes to a transcription start site are the same and the nucleotide compositions of the flanking sequences are identical, there are no differences in the effect of TATA boxes on transcription, in contrast to promoters of real genes [97]; further research on this topic is needed.

How the interactions of TBP with TATA boxes differ and how much these interactions correlate with transcriptional efficiency remains unknown, although this information may be important for the design of superpromoters.

Genome-wide studies on light-regulated gene expression during *A. thaliana* seedling development suggest that light modulates many pathways of plant growth and development [99,100]. Kiran et al. [101] have researched the impact of point mutations in a 13-nucleotide prototype sequence of a TATA box (A_1_C_2_A_3_C_4_T_5_A_6_T_7_A_8_T_9_A_10_T_11_A_12_G_13_) [102] on light-dependent transcription and noted that the substitution of T_7_ or A_8_ with G/C has almost no effect on expression in the dark, regardless of whether the light-sensitive element Ire is present (Table 6). In contrast, under light, these mutations completely inhibited transcription. Ire accomplishes photosensitive activation of transcription even from heterologous promoters.

Experiments with nuclear proteins from seedlings grown in the dark or under light have shown that sequence variations within a TATA box regulate the assembly of alternative transcription complexes [102]. The same authors [102] assayed the effect of T_9_ substitutions on the expression of the minimal *Pmec* promoter and the same TATA box prototype sequence was cloned into the pBI101 binary vector containing the *gusA* reporter gene. Tobacco leaves were transformed with the resulting constructs. The impact on transcription was evaluated in the presence of various sequences of an activator located upstream of the transcription start site. In contrast to mutations at the seventh and eighth positions [103], mutations G and C at the ninth position led to an increase in transcription under light: G_9_ by eight–nine-fold and C_9_ by three–four-fold compared with T_9_. In the dark, the expression was greater approximately five- and three-fold, respectively. 

Without the activator, the transgenic *Pmec* lines with the G_9_ mutation in the TATA box manifested more than 22-fold stronger expression compared with the transgenic *Pmec* lines having the TATA box prototype. From their results, those authors [102] concluded that the mutation of T_9_ to G or C in the TATA box gives more favorable sequence architecture, which enhances transcription. The same authors [102] also revealed that mutations G_9_ and C_9_ give rise to multiple transcription start sites, in contrast to the construct containing the prototype A_1_C_2_A_3_C_4_T_5_A_6_T_7_A_8_T_9_A_10_T_11_A_12_G_13_, with which only one transcription start site was observed [103]. From the data obtained in Ref. [102], a reasonable conclusion was made that multiple transcription start sites in the mutants may arise, because alterations of the sequence of the TATA box can result in different compositions of the preinitiation complex formed on the core promoter and, as a consequence, can cause a transcription change. This work [102] also indicates that the flanking sequences of the TATA box are important for the stability of the preinitiation complex in vivo and for gene expression specificity. By means of a rice cell extract, it has been found [104] that the replacement of TATA box sequence TATTTAA with either TCGTTAA or TATGGAA induces almost complete inactivation of the *PAL* promoter.

With the help of genome-wide microarrays of *A. thaliana*, the authors of Ref. [104] have examined the reaction of *Arabidopsis* to an attack by various insect species (caterpillars and aphids). Insects caused various changes in the expression of genes associated with amino acid metabolism. In response to the caterpillar attack, there was activation of defense gene coding for proteins involved in biosynthesis as well as activation of glucosinolate biosynthesis. Of note, the biochemical reactions seemed to be specific to herbivorous insect species; some insects suppressed or did not induce defense responses in the plants. It was shown in that report that the set of caterpillar-activated genes is enriched with the TATA boxes that may play a part in the regulation of the biotic-stress response (to rodents, caterpillars, and other pests). Genes devoid of TATA boxes perform housekeeping functions and probably do not require much transcriptional regulation.

Some researchers [105] have conducted a massively parallel reporter analysis (MPRA) to measure the activity of almost complete sets of promoters from *Arabidopsis*, maize, and sorghum. They found that core promoters containing a TATA box are four times stronger than promoters without a TATA box, especially when a TATA box is located in the region 23 to 59 bp upstream of a transcription start site, where most TATA boxes are situated in the promoters of *Arabidopsis*, maize, and sorghum. The substitution of one or two T nucleotides with G in the main TATA motif lowered transcriptional activity. Similarly, promoter strength increased when the canonical TATA box was inserted into a promoter devoid of TATA. The mutated version of the TATA box did not have this effect. To determine which promoter characteristics affect sensitivity to an enhancer, those authors analyzed the elements that influence promoter strength and found that promoters with a TATA box were 67% more sensitive to the enhancer nucleotide sequence than promoters without a TATA box.

As a consequence of the development of the Earth’s aerobic atmosphere, the need for iron in plants has risen and its deficit has become a limiting factor for plant growth. It is reported in Ref. [106] that *Arabidopsis* plants with a knockout of the *IRT1* gene (iron-regulatory transporter 1) encoding a transporter of Fe (and of zinc, manganese, cadmium, and cobalt) suffer from chlorosis and die. By modifying the promoter of the *IRT1* transporter gene, namely by inserting an additional TATA box, other investigators [107] have shown that apple trees (genus *Malus*) adapt to a deficit of iron ions (that play an important role in a wide range of cellular processes, including photosynthesis and respiration). This modification raised the gene expression and Fe uptake. To further characterize the influence of the TATA box insertion on the activity of the *IRT1* promoter, the authors of Ref. [107] assayed transient expression in *Arabidopsis* protoplasts via the addition of one to three TATA boxes fused to a *GUS* reporter. Gene expression was measured relative to GFP (green fluorescent protein) expression driven by the constitutive 35S promoter. The results uncovered a clear-cut positive correlation between the number of TATA box inserts and promoter activity. The binding of transcription factor TFIID to the *IRT1* promoter was enhanced by the presence of the TATA box insert, yielding a greater amount of *IRT1* transcripts and a higher Fe uptake. Those authors [107] believe that this is a new mechanism of genetic divergence, where an insertion of a TATA box promotes adaptation to the environment.

Different categories of genes (according to their participation in cellular processes) have dissimilar frequencies of TATA boxes [108]; for example, 60% of genes associated with oxidative stress (e.g., in *A. thaliana*) contain a TATA box in contrast to genes related to protein folding, where only 20% contain a TATA box. Those authors theorize that the presence of a TATA box in a promoter creates a conformation for a transcription factor that promotes precise and rapid transcription initiation [109]. This is important because a stress-related protein must be synthesized quickly in response to a stress signal. It was found in Ref. [81] that promoter regions of tissue-specific genes are enriched with TATA box and Y-patch motifs. Many research articles have shown that the structural properties of a promoter (to which core promoter elements contribute) modulate gene expression [110,111], e.g., through the differential binding of transcription factors.

Thus, we can say the following about the main functions of the TATA box: the overall magnitude of expression of TATA-containing genes correlates well with the sequence of the TATA box (the closer it is to the canonical one, the higher the expression) [112,113]; most variants in the TATA box consensus sequence reduce gene expression in vitro and in vivo [114,115,116,117]; the sequence of the TATA box and that of its flanking regions can affect the composition of the preinitiation complex emerging on the core promoter; the insertion of additional TATA boxes may be an auxiliary mechanism of plant adaptation to environmental conditions; the TATA box is indirectly involved in the specific regulation of the expression of light-dependent genes in plants.

## 7. Structural and Functional Roles of *A. thaliana* TBP1 and TBP2

*Arabidopsis* is known to have two genes encoding TBP [97,118,119]; the protein products are very similar in amino acid sequence to TBP1 and TBP2 of maize [120] and potato [121]. In *Arabidopsis*, these isoforms differ by 10% and may have major differences in protein-binding surfaces [24]. The examination of full-length TBP of *A. thaliana* [24] by X-ray crystallographic analysis suggests that the carboxyterminal domain of eukaryotic TBP has a pseudo-symmetrical structure formed by ten β-folds (S1–S5 and S1′–S5′, which constitute a slightly curved antiparallel β-sheet forming a concave DNA-binding saddle) and by four α-helices (H1, H2, H1′, and H2′, which lie on the upper side of the molecule). A comparison of amino acid sequences at a resolution of 2.6 Å revealed two positions that distinguish TBP1 and TBP2 in *Arabidopsis*. In TBP1, position 193 contains arginine in *Arabidopsis*, alanine in maize, and serine in both TBP2s. Position 198 features valine in both TBP1s and isoleucine in both TBP2s. This divergence may lead to isoform-specific functional differences in interactions with other proteins [24].

A comparison of protein sequences suggests that, among eukaryotic organisms, the TBP–TFIIB interaction is highly conserved, involving eight amino acid residues of TBP and 12 of TFIIB [24]. Four of the eight residues of TBP are situated in a C-terminal stirrup, whose structure is identical among the following species: *Arabidopsis* [122], humans [123], *Drosophila* [124], and yeast [125]. Of these amino acid residues, E146 (*A. thaliana* TBP2) contacts TFIIB the most by forming a firm salt bridge and two hydrogen bonds. All 12 residues in the conserved core of human TFIIB that bind to TBP are conserved in *Drosophila*, 10 are conserved in *A. thaliana*, and 8 in yeast [126]. The dependence of activated transcription on the TBP–TFIIB interaction in vivo appears to differ among species (humans, yeast, and plants). Yeast cells do not require a TBP–TFIIB interaction for transcription because it is activated by various acidic activation domains, including VP16 [127]. In HeLa cells, there is a varied dependence pattern [128] and in plants the picture is more complicated. The dependence on the TBP–TFIIB interaction varies among different activation domains and among promoters (e.g., the synthetic *Gal4* promoter versus more complex natural promoters); furthermore, the two stirrup glutamine residues of TBP differ in importance, with E146 being more crucial than E144 [129].

The upregulation of TBP2 has been detected in apical shoots of *A. thaliana*, indicating that an elevated concentration of this protein can impair cell differentiation in the leaf embryo or in young shoots, thereby turning them into meristem-like structures [130]. Overexpression of *TBP2* in transgenic *A. thaliana* plants induces proliferation of apical shoots and the expression of some genes (*STM* and *KNAT1*) regulating the shoot meristem undergoes changes. All these findings suggest that the expression level of the *TBP* gene may play a major part in the control of shoot formation and plant morphology [131]. 

In flowering plants, spermatozoa are delivered to the embryo sac by a pollen tube guided by female signals. Both gametic and synergid cells contribute to the guiding of pollen tubes. Synergids secrete peptide signals that anchor the tube, whereas the function of gametic cells is unknown. Some investigators [130] have identified CCG-binding protein 1 (CBP1) and proved that it interacts with CCG (central cell guidance, required to attract pollen tubes in *A. thaliana*), with mediator subunits, with Pol II, and with central-cell-specific AGAMOUS-like transcription factors. Additionally, CCG interacts with TBP1 and Pol II as a TFIIB-like transcription factor. CBP1 is expressed in many vegetative tissues, especially in the central cell during reproductive growth. Some scientists [130] believe that CBP1, via an interaction with CCG and the mediator complex, binds to transcription factors and Pol II machinery in order to modulate the attraction of pollen tubes. CBP1 is conserved among plants and shares no homology with animal or yeast proteins.

The authors of Ref. [97] have documented functional differences between two isoforms of TBP of *Arabidopsis* (TBP1 and TBP2) by means of three types of promoters: from the *U2* gene of small nuclear RNA and from mRNA genes read by Pol II as well as from the *U6* gene of small nuclear RNA generated by RNA polymerase III. Transcriptional conditions for a yeast system [97] were employed for this purpose, because the C-terminal domains of all three TBPs are highly conserved (a 10% difference in amino acid sequence). Mutant TBP1 and TBP2 that carry three amino acid substitutions in the second direct repeat of TBP manifested aberrant specificity of binding to DNA and recognized not only a mutant TATA box having sequence TGTAAA but also other variants of the TATA box, in contrast to yeast TBP [132]. Expression from the *U2* gene was four-fold higher with the mutant TATA box TGTAAA and two-fold higher with mutant TATA boxes, in which TATAAA was replaced with TCTAAA, TAGAAA, or TATAGA. Mutant human TBP also worked with TGTAAA (a mutant sequence) in all three gene types [97]. Therefore, both *Arabidopsis* isoforms of TBP are functionally identical if we compare transcriptional systems of Pol II and III in plant cells.

In *Arabidopsis*, as in other eukaryotes, TBP and TBP-associated factor (TAF)-like proteins have been identified [118,133,134]. An extensive work using a yeast two-hybrid system to map interactions of putative components of *A. thaliana* TFIID [135] has shown that plant TFIID may have some unique properties. For example, *Arabidopsis* possesses two isoforms of TAF1. One isoform of TAF1 has a domain structure homologous to that of other eukaryotes, while the second isoform differs from the first one in that its N-terminal domain is shorter by 91 amino acid residues [135].

In that study, among several TBP-associated factors of this plant (TAF9, TAF10, TAF11, and TAF12), none could serve as a substitute for their yeast homologs. An analysis [36,136] has revealed that both *Arabidopsis* isoforms of TBP interact only with the N terminus of TAF1, the portion of the protein predicted to contain the N-terminal domain that binds to TBP in yeast and *Drosophila*.

## 8. Participation of TBP in Other Cellular Processes

The transcriptional activation of genes encoding heat shock proteins (HSPs) is a highly conserved response to heat and other environmental stressors in all organisms [137,138,139]. The response to heat stress is critical to the life cycle and yield of plants, especially food crops grown in the field. A comparison of heat shock protein sequences among different organisms points to the presence of a conserved DNA-binding domain and three hydrophobic heptapeptide repeats known as the trimerization domain. These domains are localized to the N-terminal portion of the heat shock protein molecule. The transcriptional activation domain is located closer to the C-terminal domain of the molecule. Intramolecular interactions between the N- and C-terminal domains keep Hsf1 in an inactive state in the absence of stress [137]. Some scientists [140] have assessed the ability of *A. thaliana* HSF1 to interact with two recombinant TBPs (TBP1 and TBP2) by affinity chromatography and an electrophoretic mobility shift assay and have uncovered the formation of complexes HSF1–TBP in vitro. In this context, TBP1 binds to HSF1 with higher affinity than TBP2 does [140]. By means of a yeast two-hybrid system, those authors proved that HSF1 and TBP1 also interact in vivo. It is likely that the interaction of TBP with HSF promotes the assembly of the remaining protein subunits of the preinitiation complex. This notion is supported by the binding of TBP to a *Drosophila* heat shock gene promoter in the absence of stress [137,141].

TBP not only performs an important function in the regulation of transcription but is also implicated in the control of transcription-related repair, which ensures efficient removal of DNA damage from transcribed genes [142]. Bacterial virulence protein VirD2 is key to the nuclear import and chromosomal integration of DNA carried by *Agrobacterium* into different organisms. VirD2 also interacts with TBP in vitro and in *Agrobacterium*-transformed *Arabidopsis* cells [143,144]; an assay of protein interactions by SDS-PAGE revealed that GST-VirD2 interacts with His_6_-TBP.

TBP recognizes only supercoiled DNA structures similar to those seen in the TATA box when bound to TBP [144]. Several transcription activators (proteins), including VP16, the adenoviral E1A protein, the HIV-1 transactivator protein Tat, and p53 also interact with a TBP [145,146,147,148] that recognizes specific DNA lesions having a structure similar to that of a TATA box complexed with TBP.

Some research [144,149] indicates that TBP can bind to DNA damaged by cisplatin or UV radiation, which are used in the treatment of cancer. Computational 3D structural analysis has detected a surprising similarity between the structure of a TATA box complexed with TBP and the structure of oligonucleotides damaged by UV radiation or cisplatin. A binding analysis based on nitrocellulose filters and DNase I footprinting revealed that damaged DNA can serve as a competitive binding site for TBP, thereby distracting it from the promoter’s TATA box and thus delaying nucleotide excision repair, inhibiting RNA synthesis, and causing tumor cell death during cancer treatment.

Some reports [150,151] indicate that TBP–TATA affinity and sequences flanking a TATA box affect the functions of Mot1 (modifier of transcription 1) and of NC2 (negative cofactor 2) in vivo. Mot1 regulates transcription, together with other cofactors, by influencing the genomic distribution and activity of TBP [152,153]. Hydrolysis of ATP is needed to dislodge TBP from the *Mot1* promoter [153]. Mot1 acts as a transcriptional repressor on stress-sensitive genes that tend to contain high-affinity binding sites for TBP in their promoters [154,155]. Conversely, Mot1 serves as a transcription activator on promoters carrying weak TATA boxes [156,157]. Mot1 competes for TBP binding with other factors, including TFIIA, TAF1, and others [158,159]. In vivo functions of NC2 and Mot1 overlap [160,161]. NC2 stabilizes the binding of Mot1 to DNA in the absence of ATP, but Mot1 can dissociate complexes with or without the help of NC2 [162]. NC2 represses transcription from various yeast, mammalian, and viral promoters. This observation implies that the function of TBP in the transcription of class II genes is regulated by including but not limited to diverse multi-subunit complexes. Much evidence suggests that the activity and accessibility of TBP can be modulated to accurately control gene transcription and that these properties can play an important role in cell type regulation.

It is known that during mitosis, when chromatin condenses, transcription is stopped and then regulated in a cell-cycle specific manner. Most transcription factors remain associated with mitotic chromosomes, but in a dynamic manner. Ref. [163] suggests that TBP (and other transcription factors) remains globally associated with active promoters in mouse embryonic stem cells during mitosis and promotes the recruitment and binding of Pol II to mitotic chromosomes, thereby facilitating transcriptional reactivation after mitosis. This mechanism represents an efficient way for nascent cells to rapidly revert to a gene activity pattern appropriate for their cell type.

It must be mentioned that, because of the high homology of a C-terminal region to that of mammalian TBP, proteins related to TBP have been identified: TBPL1 (TBP-like 1, also known as TRF2 (TBP-related factor 2)), TLP, TBPL2, and TRF3. A good review of this topic is given in Ref. [164].

## 9. Conclusions

Unlike most animals, plants are sedentary and cannot exit a poor-quality environment after germination. Consequently, in order to survive, they need to adapt to the specific conditions they face. In order to adapt to rapidly changing conditions, cells often regulate gene expression at the transcriptional level. At this level, elements of the TATA box affect the recruitment of cofactors and transcription factors to promoter regions of target genes. Changes in the amount of TBP in *A. thaliana* participate in plant development by affecting the organization of shoot apical meristems, leaf development, flower organ and leaf formation, fertility, pollen tube growth, and the light response. Promoters of genes containing TATA boxes are occupied by the regulatory molecules that are different from those that bind to the promoters of TATA-free genes; this arrangement greatly contributes to differences in responses to internal and external signals. This means that differences in specific elements of the promoter region, in particular in the TATA box, may have dissimilar functional effects on the cell [165]. Therefore, a comprehensive understanding of the *A. thaliana* genome and accurate insights into its gene expression are of great importance for both basic research and practical applications. The architecture and composition of core promoters containing the TATA box are highly dependent on the nucleotide sequence of the TATA box and on the nucleotide composition of the sequences flanking the TATA box. A large number of studies have addressed the interaction of TBP with TATA boxes and TATA-like elements in mammals and this number has begun to increase with the discovery of the influence of single-nucleotide polymorphisms in TATA sequences on phenotypic manifestations, in particular, correlations with various human diseases. Research into the interaction of TBP with TATA boxes of promoters of real plant genes, in particular *A. thaliana* genes, is noticeably lagging. The results of such studies will be useful for practical applications and should improve our understanding of transcription initiation in plants.

## Figures and Tables

**Figure 1 plants-12-01000-f001:**
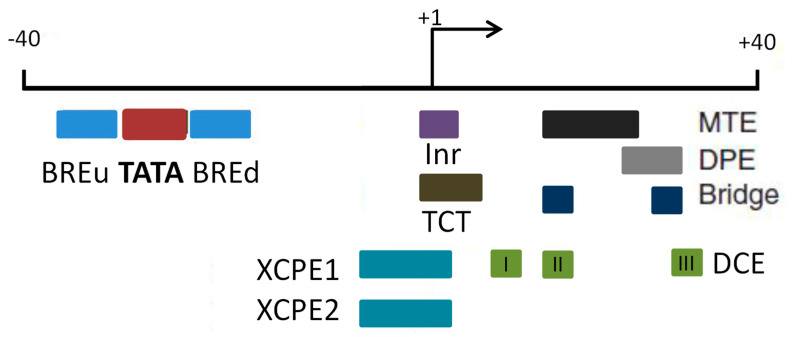
Some core promoter elements taking part in transcription by Pol II. Motifs termed BREu, BREd, TATA box, Inr, MTE, DPE, and TCT (polypyrimidine initiator), and the TCT motif are thought to be present in nearly all ribosomal protein gene promoters in *Drosophila* and humans [17].

**Figure 2 plants-12-01000-f002:**
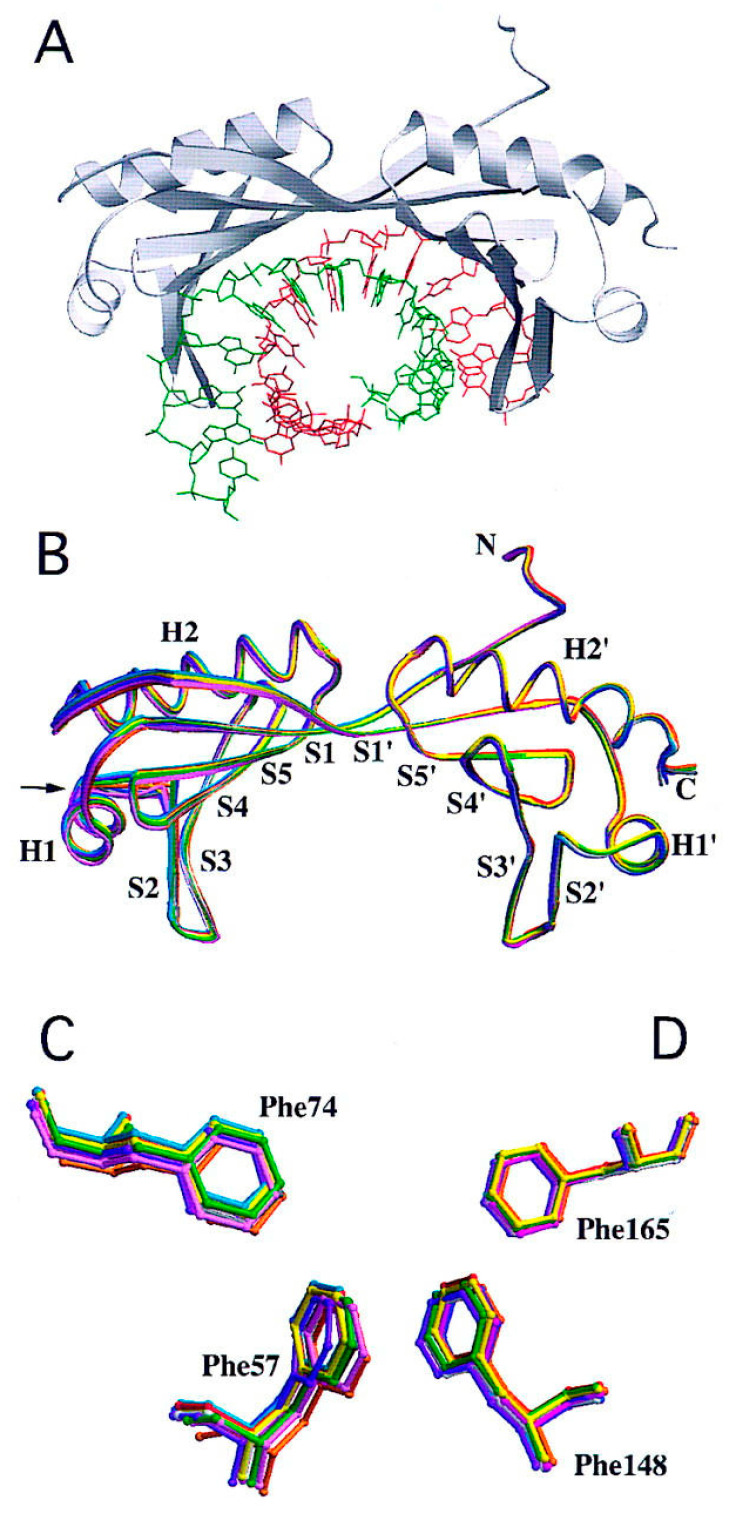
Structure of the TBP–TATA complex. (**A**) TBP in the complex with a TATA box of *AdMLP*. The coding strand of DNA is green; the helix and strands of TBP are grey. (**B**) Overlay of the polypeptide backbones of 11 TBP–DNA complexes, by use of a spaghetti representation with a different color for each cocrystal structure and the view shown in A. Atomic stick figure representations of the amino-terminal (Phe-57 and Phe-74) (**C**) A superposition of several polypeptide backbones in a TBP–TATA complex. Panels (**C**,**D**) are atomic stick figures of N-terminal (Phe-57 and Phe-74) and C-terminal (Phe-148 and Phe-165) phenylalanine pairs from TBP–DNA complexes [33].

**Figure 3 plants-12-01000-f003:**
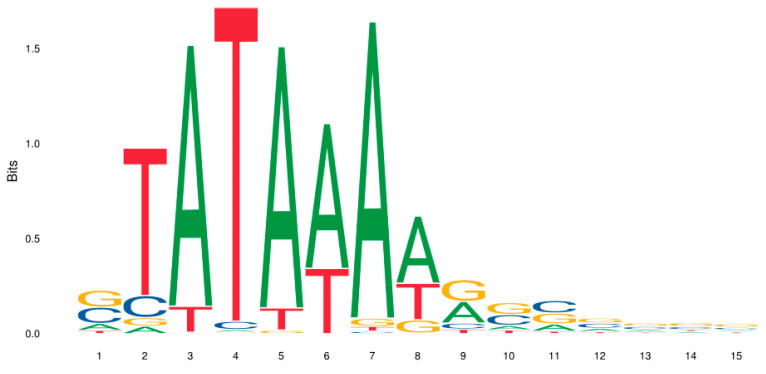
A frequency matrix for a TATA box [57] presented as a sequence logo.

**Figure 4 plants-12-01000-f004:**
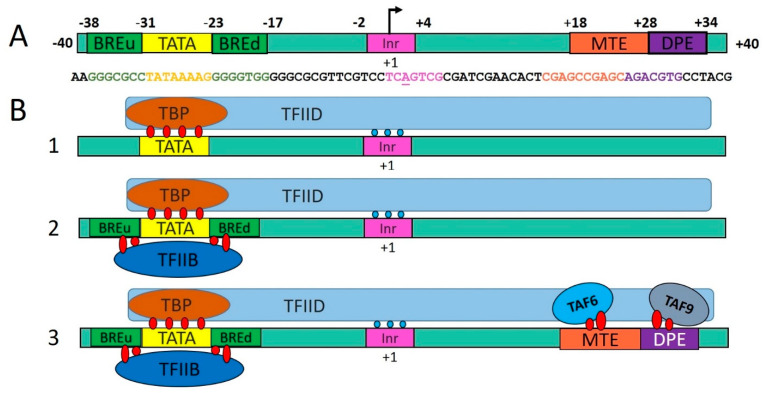
The scheme of DNA–protein contacts arising during the interactions of DNA-binding general transcription factors with a core promoter [5,17]. (**A**) The scheme of location of some core promoter elements on DNA; (**B**) 1: formation of the TBP-TATA complex; 2: interaction of TFIIB with BRE_u_ and BRE_d_, formation of the TFIIB-TBP-TATA complex via protein–protein and protein–DNA interaction. Although there are only two general transcription factors interacting with DNA (TFIID and TFIIB), they can engage in many contacts with the core promoters and can interact with each other; 3: the interaction of TAF6 and TAF9 TFIID subunits with MTE (motive ten elements) and DPE (downstream promoter element) core promoter elements, which are enriched in TATA-free promoters, is shown.

**Figure 5 plants-12-01000-f005:**
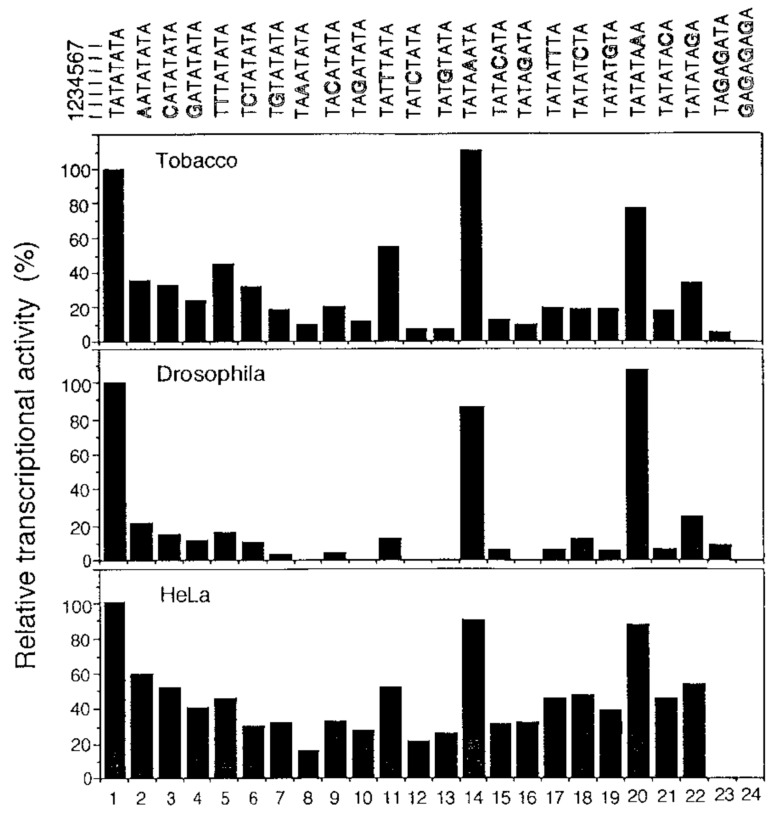
Transcriptional efficiency levels seen with templates containing various mutated *TC7* TATA elements in tobacco, *Drosophila*, and HeLa system [98].

**Table 1 plants-12-01000-t001:** A frequency matrix for a TATA box [58].

	−3	−2	−1	0	1	2	3	4	5	6	7	8	9	10	11
A	61	16	352	3	354	268	360	222	155	56	83	82	82	68	77
C	145	46	0	10	0	0	3	2	44	135	147	127	118	107	101
G	152	18	2	2	5	0	20	44	157	150	128	128	128	139	140
T	31	309	35	374	30	121	6	121	33	48	31	52	61	75	71
	G	T	A	T	A	A	A	A	G	G	C	G	C	G	G
	C		T		T	T		T	A	C	G	C	C	C	C

**Table 2 plants-12-01000-t002:** The nucleotide frequency matrix for the TATA box from 171 unrelated plant promoters.

	<2	<1	1	2	3	4	5	6	7	8	>1	>2
A	0.28	0.16	0.03	0.95	0.00	1.00	0.62	0.97	0.38	0.73	0.13	0.30
C	0.27	0.63	0.01	0.00	0.04	0.00	0.00	0.00	0.01	0.08	0.42	0.42
G	0.17	0.05	0.00	0.00	0.00	0.00	0.00	0.02	0.00	0.10	0.28	0.16
T	0.28	0.16	0.96	0.05	0.96	0.00	0.38	0.01	0.61	0.09	0.18	0.11
		c	T	A	T	A	A/T	A	T/A	A		

**Table 3 plants-12-01000-t003:** Effects of substitutions in *A. thaliana* TATA boxes on in vitro transcription in a *HeLa* nuclear extract [96].

TATA Box	Expression, %
TATATATA	100
TAAATATA	15
AATATATA	>36
TATAAATA	>36
TATATAAA	>36
TTTATATA	>36
TATTTATA	>36
TATATTTA	>36
TATATATT	>36
TAGAGATA	0
GAGAGAGA	0

**Table 4 plants-12-01000-t004:** The influence of mutations in a TATA box on mRNA expression of the *CaMVsynT-3* gene in *A. thaliana* protoplasts [97].

TATA Box	Expression, %
TATAAATA	100
TACGAATA	5
TATACGTA	5
CGTAAATA	7
TATAAACG	27

**Table 5 plants-12-01000-t005:** Rough quantitative estimates of transcription efficiency in three eukaryotic species for different mutations of TATA boxes [98].

#	TATA Box	Tobacco, %	*Drosophila*, %	*HeLa*, %
1	TATATATA	100	100	100
2	AATATATA	<36	~20	<60
3	CATATATA	<36	~17	<50
4	GATATATA	~20	~10	~40
5	TTTATATA	~40	~15	>40
6	TCTATATA	>30	~10	~30
7	TGTATATA	<20	<5	~30
8	TAAATATA	~7	0	~17
9	TACATATA	>10	<5	~30
10	TAGATATA	~7	0	>30
11	TATTTATA	>40	~10	<60
12	TATCTATA	<5	0	>20
13	TATGTATA	<5	0	~30
14	TATAAATA	112	<70	~90
15	TATACATA	~5	<7	<40
16	TATAGATA	<5	0	<40
17	TATATTTA	>7	<7	~50
18	TATATCTA	>7	~10	~50
19	TATATGTA	>7	<7	~40
20	TATATAAA	~70	~110	~90
21	TATATACA	>7	<7	~50
22	TATATAGA	~30	~25	~57
23	TAGAGATA	0	<10	0
24	GAGAGAGA	0	0	0

**Table 6 plants-12-01000-t006:** Relative amounts of gusA transcripts in the dark and under light after mutations in the TATA box [101].

Promoter Mutation	Dark	Light
P_mec_	1.04 ± 0.20	1.00
T_7_→C	0.79 ± 0.16	0.01 ± 0.00
T_7_→G	2.04 ± 0.40	0.02 ± 0.00
A_8_→C	0.42 ± 0.08	0.01 ± 0.00
A_8_→G	1.68 ± 0.32	0.02 ± 0.00

## Data Availability

Data available in a publicly accessible repository. The submitted manuscript complies with the editorial and ethical policy of MDPI.

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
