# Peer review of "On the Role of TATA Boxes and TATA-Binding Protein in Arabidopsis thaliana"

_plants, 2023, doi:10.3390/plants12051000_

Round 1

Reviewer 1 Report (New Reviewer)

This manuscript is a good review of the TATA box and its variants in Arabidopsis. It also describes the differences between TBP1 and TBP2 and discusses their putative roles in the stress response. It is well written. Some indications in the draft have been included and must be answered before the final acceptance of the manuscript.

Author Response

Thank you very much for your comments. Please see the attachment

Reviewer 2 Report (Previous Reviewer 2)

The revisions made are not satisfactory. The language needs extensive editing. data is also poorly organized in tables.

Author Response

Thank you very much. Good. The manuscript was given for language checking by a certified translator.

Reviewer 3 Report (New Reviewer)

In this manuscript, the authors describe the molecular mechanism of TATA boxes and TATA-binding protein and their involvement in morphology development in Arabidopsis thaliana, which is an interesting topic. Some issues need to be addressed.

1)The resolution of the Figures in this manuscript is too low.

2)The contents of 2. Structure of TBP, and 3. Structure and variation of the TATA box should change the order according to the article's structure.

Author Response

 Thank you.  Good. We believe that the order of sections 2 and 3 corresponds to the              structure of the article.

Reviewer 4 Report (New Reviewer)

Many figures in this review (Fig 2,4,5,6) were directly copied from the reference, which you may need to get permission from authors/publisher. The author should make their own summary figures to give the audience more insight into this field.

Tables 1 & 2 are not coherent. For example, table 1 has site 0 of TATA-box, but table 2 starts from 1. There are -3~-1 sites in table 1, while they were labeled as <2 <1 in table 2. The definition of the sites need to be consistent. 

Author Response

Thank you. We have obtained permission from all authors to use the drawings. Tables 1 and 2 are presented in the author’s edition, which allows us to draw the necessary conclusions. We have adapted Figure 4 to our text. We have inserted a ref. to Baptista:

"Baptista et al. (68) have re-examined the participation of SAGA in global transcription in S. cerevisiae and demonstrated that it is the general transcription factor that is recruited to most promoters of genes transcribed by Pol II, where SAGA plays a critical role in mRNA synthesis regardless of the promoter type (SAGA- or TFIID-dominated, TATA-containing, or TATA-free). In this regard, SAGA can be compared to Mediator, which is required for the whole transcription by Pol II and stimulates the formation of the preinitiation complex. The authors of ref. (68) concluded that the question about the function of SAGA in transcription in higher eukaryotes is still open. In higher eukaryotes, as a rule, TATA-containing (and, accordingly, TFIID-dependent) genes are inducible not housekeeping genes. It is very likely that in higher eukaryotes, the regulation of transcription of genes directed by TATA-like elements also has its own specific features. There is currently no validated definition of a TATA-like element in higher eukaryotes."

Round 2

Reviewer 3 Report (New Reviewer)

Almost all queries that need to be addressed in the revised manuscript have been modified carefully by L.K. Savinkova and colleagues, I think this manuscript can be accepted.

Author Response

Thank you for your review

This manuscript is a resubmission of an earlier submission. The following is a list of the peer review reports and author responses from that submission.

Round 1

Reviewer 1 Report

This is a carefully presented review reporting the role of TATA boxes and TATA-binding protein in Arabidopsis thaliana.  The data presented are up-to-date and this review will be interesting to readers of "Plants", since this kind of work is missing from the literature. I suggest to be accepted for publication after moderate changes in English language.

Reviewer 3 Report

Based on the title “On the role of TATA boxes and TATA-binding protein in Arabidopsis thaliana”, I expected this review to discuss an important and not fully understood part of plant transcription: the role of the TATA box and distinct TBPs. But rather than focusing on what is suggested from the title, a large part of this review simply recites seminal reviews, and unfortunately, that not even always correctly! In general, this review lacks focus, is often wrong as statements from mammals or flies do not copy plants, could be greatly improved by a thorough and dedicated review of the literature (especially the last 5-10 years), attention to detail, and potentially, scientific editing. At times, descriptions are wrong, misleading, or unprecise. It is not clear if this is due to writing or understanding, but either way, it is potentially damaging to the field. i.e. “[..] this positioning ultimately determines the placement of Pol II relative to a TSS L84.” What does this mean? Do the authors try to say that the TBP determines position of the PIC and thus TSS? “Genes devoid of TATA boxes perform housekeeping functions and probably do not require much transcriptional regulation.” L320 – so >60% in AT? How about some PIFs and other factors that lack TATAs? While very often using conditionals, other statements like the above one appears to be states as a fact that at least to me, is barely supported by the literature. (While disperse regulation is enriched for regulating housekeeping genes, even the generous estimation that 29-39% of plant promoters contain TATAs should have made the authors wonder, are really 61-71% of A.t. genes housekeeping?)

Among a long list,

·       it should be considered to not use ‘transcription’ when referring to a particular aspect of it, i.e., transcription initiation,

·       when sentences need to be tailored and not just regurgitated (L44/389 “In eukaryotes, transcription is carried out by three DNA-dependent RNA polymerases.” - Especially in a PLANT review I think it is worth to acknowledge at least Pol IV/V)

·       Since 2003, the consensus for the Inr has been reannotated (L 59) – for example, an INR for plants is published in PMID: 27729530, the human Inr PMID: 28108474, others PMID: 36030508

·       L59: “highly common core promoter elements located” – how common is the MTE? And how common is it in plants? Same for the DPE. What is common? A % over background would be helpful to see how valid this statement is. How about the DPE?

·       L78, PMID: 8034589, consider refining to not be misleading

·       L80: how many TAFs are there in plants?

·       L88 to 92: For most of human promoters devoid of a canonical TATA box sequence (19, 20), the precise positioning of TBP is determined by subunits called TAFs, which collectively act as a molecular ruler to place TBP accurately on the core promoter. Obviously, on TATA-containing promoters, the specificity of the DNA–TBP interaction is due to 91 DNA–protein interactions, whereas on promoters without TATA, this specificity is predicated on direct and indirect interactions with activator proteins and TAFs (12). I am not sure this is a correct statement. Could the authors clarify the role of TAFs in TATA-less promoters and cite relevant literature. Do some TFs also interact with other parts of the PIC aside from the TAFs? How about TFIIA/B?

·       What are the A.t. homologues of Mot1/Nc2?

The provided figures also provide little novelty, and largely unrelated to the title. Many CPEs shown in Fig 1 are likely irrelevant for plant transcription and appears to just be redrawn. Which of these elements are found in plants? Are there elements found in plants not mentioned? i.e. the TGT motif?

Last, I refer to the editor’s judgement on this issue but would like to highlight that the authors cite many of the key reviews in the field. While no doubt, every single one of them is a celebrated, highly cited, and well visible masterpiece, I’d like to raise three concerns. The first is related to the utility of this style for readers, most of which are likely aware of these reviews. Hence little inspiration, or novelty is the likely result. Second, most of these reviews are >10 years old and themselves cite >100 papers. Does this mean nothing worth mentioning has been published since? This would undermine the need for this review. Last, aside from the fact that it is not obvious which X citations of the cited reviews provide the data, I’d like to draw awareness to the fact that reviews citing reviews exaggerate the ‘Matthew Effect’. This not only causes frustration, but also loss of diversity in both funding and thought in a field and, as seen in this review, oversight of advances including function of TBP-like proteins, discovery of new core promoter elements, reanalysis of consensus sequences using novel techniques resulting in new understanding.